# AAMP and MTSS1 Are Novel Negative Regulators of Endothelial Barrier Function Identified in a Proteomics Screen

**DOI:** 10.3390/cells13191609

**Published:** 2024-09-25

**Authors:** Fabienne Podieh, Max C. Overboom, Jaco C. Knol, Sander R. Piersma, Richard Goeij-de Haas, Thang V. Pham, Connie R. Jimenez, Peter L. Hordijk

**Affiliations:** 1Department of Physiology, Microcirculation, Amsterdam Cardiovascular Science, Amsterdam UMC, 1081 HV Amsterdam, The Netherlands; f.a.podieh@amsterdamumc.nl (F.P.);; 2Department of Medical Oncology, OncoProteomics Laboratory, Cancer Center Amsterdam, Amsterdam UMC, 1081 HV Amsterdam, The Netherlandsc.jimenez@amsterdamumc.nl (C.R.J.)

**Keywords:** ubiquitin, endothelial cells, monolayer integrity, Rho GTPases, protein turnover

## Abstract

Cell–cell adhesion in endothelial monolayers is tightly controlled and crucial for vascular integrity. Recently, we reported on the importance of fast protein turnover for maintenance of endothelial barrier function. Specifically, continuous ubiquitination and degradation of the Rho GTPase RhoB is crucial to preserve quiescent endothelial integrity. Here, we sought to identify other barrier regulators, which are characterized by a short half-life, using a proteomics approach. Following short-term inhibition of ubiquitination with E1 ligase inhibitor MLN7243 or Cullin E3 ligase inhibitor MLN4924 in primary human endothelial cells, we identified sixty significantly differentially expressed proteins. Intriguingly, our data showed that AAMP and MTSS1 are novel negative regulators of endothelial barrier function and that their turnover is tightly controlled by ubiquitination. Mechanistically, AAMP regulates the stability and activity of RhoA and RhoB, and colocalizes with F-actin and cortactin at membrane ruffles, possibly regulating F-actin dynamics. Taken together, these findings demonstrate the critical role of protein turnover of specific proteins in the regulation of endothelial barrier function, contributing to our options to target dysregulation of vascular permeability.

## 1. Introduction

Endothelial cells (ECs) form the inner lining of blood vessels and are crucial for vascular integrity, attributed to their tight control of extravasation of fluid, proteins, and leukocytes across the vessel wall. Loss of this endothelial barrier function causes (hyper)permeability and leakage of plasma proteins, leading to edema formation and tissue damage and, eventually, life-threatening conditions. Preservation of endothelial integrity is mediated by tight adhesion of vascular endothelial (VE)-cadherin-containing adherens junctions and the dynamics of the actin cytoskeleton, which is indirectly linked to the intracellular part of VE-cadherin [1,2,3,4,5].

Rho GTPases are master regulators of actin cytoskeleton dynamics and, thus, play a prominent role in endothelial barrier regulation. Rac1 and Cdc42 induce formation of a cortical actin network promoting the stabilization of junctions, whereas RhoA and RhoB drive formation of contractile F-actin stress fibers, resulting in impaired cell–cell contact [6,7,8]. Activity of Rho GTPases is controlled by their cycling between an inactive, GDP-bound state and an active, GTP-bound state [9,10]. In addition, posttranslational modifications, such as ubiquitination, further control Rho GTPase stability and downstream signaling capacity. Ubiquitination involves activation of the ubiquitin molecule by E1 ligases. Subsequently, ubiquitin is transferred to an E2 ligase and, ultimately, to E3 ligases, which catalyze the covalent attachment of the ubiquitin peptide to a lysine residue of a specific substrate. This may initiate ubiquitin chain formation by the subsequent addition of ubiquitin peptides to any of the lysines in the first and consecutive ubiquitins. For example, K48- or K63-linked ubiquitin chains target the substrate for proteasomal or lysosomal degradation, respectively [11,12]. The importance of ubiquitination as a mechanism that controls the stability of specific proteins, which in turn control (endothelial) junctions, has been shown previously [13,14]. For example, the E3 ligase CHFR targets VE-cadherin for degradation, thereby directly regulating cell–cell adhesion [15]. The contraction-inducing Rho GTPase RhoB, unlike its close homolog RhoA, is constitutively targeted for lysosomal degradation, which preserves quiescent endothelial integrity [16,17].

Recently, we showed that ubiquitin-mediated degradation of RhoB by a Cullin3-based E3 ligase complex is a crucial regulatory event to preserve endothelial barrier function, by preventing excessive accumulation of this negative regulator of barrier function [17]. Further, our data demonstrated that in the quiescent endothelium, there exists a more generic role for protein turnover in maintaining endothelial integrity. Inhibiting ubiquitination and degradation via E1 ligases leads to an immediate loss of barrier function, whereas blocking protein translation improves endothelial barrier function, suggesting that predominantly negative barrier regulators, such as RhoB, are controlled by fast protein turnover [16]. We also showed that disruption of endothelial barrier function induced by inflammatory mediators, such as TNFα and IL-1β, is controlled by protein turnover [16]. These data suggest a critical role of protein synthesis and degradation for regulating the signaling capacity of specific proteins in ECs, controlling barrier function of quiescent and inflamed endothelium.

Based on these previous observations, we sought to identify other proteins, next to RhoB, whose stability and signaling capacity is regulated by protein turnover, using a proteomics approach. Human ECs treated for 6 h with the E1 ligase inhibitor MLN7243 or the Cullin E3 ligase inhibitor MLN4924 were used to identify differentially expressed proteins. Sixty proteins were found to be significantly up- or down-regulated upon inhibition of ubiquitination. Angio-associated migratory cell protein (AAMP) and metastasis suppressor protein 1 (MTSS1), both of which were found to have a short half-life (~1–4 h), were identified as novel negative regulators of endothelial barrier function. Mechanistically, AAMP controls both RhoA and RhoB activity and stability. Further, our data suggest that AAMP colocalizes with F-actin and cortactin at membrane ruffles, suggesting that AAMP is involved in regulating F-actin dynamics, directly or indirectly via Rho GTPases. Taken together, we show that rapid and continuous protein turnover of negative regulators, such as AAMP and MTSS1, plays a critical role in maintaining endothelial integrity.

## 2. Material and Methods

### 2.1. Antibodies and Reagents

For Western blot analysis, the following primary antibodies were used: anti-AAMP (#PA5-31454, Invitrogen, Waltham, MA, USA), anti-MTSS1 (#93065, Cell Signaling Technology, Danvers, MA, USA), anti-RhoB (#14326-1-AP, Proteintech, Manchester, UK), anit-CLASP1 (#ab108620, Abcam, Cambridge, UK), anti-SQSTM1/p62 (#5114, Cell Signaling Technology), anti-GAPDH (#2118, Cell Signaling Technology), anti-VE-cadherin (#2500, Cell Signaling Technology), anti-Rac1 (#610650, BD Transduction Laboratories, Franklin Lakes, NJ, USA), anti-RhoA (#2117, Cell Signaling Technology), anti-Cdc42 (#2466, Cell Signaling Technology), anti-pMLC (#3671, Cell Signaling Technology), anti-myc (#2276, Cell Signaling Technology), anti-cortactin (#05-180-I, Merck, Darmstadt, Germany), anti-p34-Arc (#07-227-I, Merck), and anti-Vinculin (#V9131, Sigma-Aldrich, St. Louis, MO, USA). As secondary antibodies, HRP-conjugated goat anti-rabbit and anti-mouse antibodies (Dako, Glostrup, Denmark) were used.

For immunofluorescent staining, the following primary antibodies were used: anti-VE-cadherin (#2500, Cell Signaling Technology and #MAB9381, R&D systems, Minneapolis, MN, USA), anti-pMLC (#3671, Cell Signaling Technology), anti-myc (#2276 and #2278, Cell Signaling Technology), anti-cortactin (#05-180-I, Merck), and anti-p34-Arc (#07-227-I, Merck). Alexa-555 donkey anti-rabbit, Alexa-647 donkey anti-mouse, Alexa-488 donkey anti-mouse, Alexa-555 donkey anti-mouse, and Alexa-647 donkey anti-rabbit (Invitrogen) were used as secondary antibodies. The nucleus was stained with DAPI (#62248, Thermo Fisher Scientific, Waltham, MA, USA) and F-actin with Acti-stain 670 phalloidin (#PHDN1-A, Cytoskeleton, Denver, CO, USA).

In this study, the compounds MLN7243 (#8341, Selleck Chemicals, Houston, TX, USA), MLN4924 (#S7109, Selleck Chemicals), and Cycloheximide (#239764, Sigma-Aldrich) were used.

### 2.2. Cell Culture

Primary human umbilical vein endothelial cells (HUVECs; #CC-2519, Lonza, Basel, Switzerland; #C-12203, PromoCell, Heidelberg, Germany) were cultured with Endothelial Cell Medium (ScienCell Research Laboratories, Carlsbad, CA, USA) on fibronectin-coated plates. The medium was refreshed every second day. The cells were cultured at 37 °C in 5% CO_2_ and used for experiments until passage 4.

HEK293T cells (#CRL-3216, ATCC) were cultured in Dulbecco’s Modified Eagle Medium (#31966–021, Gibco, Waltham, MA, USA) supplemented with 100 U/mL penicillin and streptomycin (Gibco), 2 mM L-glutamine (Capricorn Scientific, Ebsdorfergrund, Germany), 1 mM sodium pyruvate (#11360–070, Gibco), and 10% heat-inactivated Fetal Bovine Serum (#A15-101, PAA, Linz, Austria). The cells were cultured at 37 °C in 5% CO_2_.

### 2.3. Sample Preparation for Proteomics

HUVECs in passages 2–3 were seeded on fibronectin-coated 21 cm^2^ dishes. When confluency was reached, cells were treated with 0.005% DMSO as a carrier control, 500 nM MLN7243, or 500 nM MLN4924 for 6 h. HUVECs were lysed with lysis buffer containing 10 mM Tris-HCl pH 7.5, 150 mM NaCl, 0.5 mM EDTA, 0.5% NP40, and 0.09% NaN_3_ in MilliQ (Merck), with freshly added 1 mM PMSF and protease inhibitor cocktail (#5871, Cell Signaling Technology). After centrifugation of lysates at 1400 rpm and 4 °C for 10 min, the protein concentration of the supernatant was determined using a BCA protein assay kit (#23227, Thermo Fisher Scientific, Waltham, MA, USA). We performed three independent experiments, leading to samples in triplicates per condition. To check for good sample quality, 10 µg of protein was mixed with 2× SDS sample buffer (125 mM Tris-HCl pH 6.8, 4% SDS, 20% glycerol, 100 mM DTT, and 0.02% Bromophenol Blue in MilliQ) and separated by SDS-PAGE by using a 4–15% Mini-PROTEAN TGX Precast gel (BioRad, Hercules, CA, USA). The gel was fixed in a solution of 50% ethanol and 3% phosphoric acid in MilliQ and stained with 0.1% Coomassie Brilliant Blue G-250 solution containing 34% methanol, 3% phosphoric acid, and 15% ammonium sulfate in MilliQ, followed by destaining in MilliQ. The image of the gel is provided in Appendix A.

### 2.4. Proteomics

#### 2.4.1. In-Gel Digestion

Cell lysates were loaded on a precast 4–12% NuPAGE SDS-PAGE gradient gel (Thermo Fisher Scientific), and gel electrophoresis was stopped as soon as the protein content had entered the running gel as a single broad band (“blob”). Proteins were reduced and alkylated in-gel by incubating in 10 mM dithiothreitol/50 mM ammonium bicarbonate (1 h, 56 °C) and 54 mM iodoacetamide/50 mM ammonium bicarbonate (45 min in the dark), respectively. Per sample, blobs were cut into ~1 mm^3^ cubes, transferred to an Eppendorf tube, and incubated with 6 ng/mL of Sequencing Grade Modified Trypsin (Promega, Leiden, The Netherlands) in 50 mM ammonium bicarbonate (O/N, 25 °C). Tryptic peptides were recovered from gel cubes in three 100–150 µL extraction rounds (once in 1% formic acid and twice in 5% formic acid/50% acetonitrile). The extracts were combined and stored at −20 °C until LC-MS/MS analysis.

#### 2.4.2. NanoLC-MS/MS

Peptides were separated by an Ultimate 3000 nanoLC system (Dionex LC-Packings, Amsterdam, The Netherlands) equipped with a 40 cm × −75 μm ID fused silica column custom packed with 1.9 μm of 120-Å ReproSil Pur C18 aqua (Dr Maisch GMBH, Ammerbuch-Entringen, Germany). After injection, peptides were trapped at 6 μL/min on a 1 cm × 100 μm ID trap column packed with 5 μm of 120-Å ReproSil C18 aqua at 2% buffer B (buffer A: 0.05% formic acid; buffer B: 80% acetonitrile/0.05% formic acid) and separated at 300 nL/min in a 10–40% buffer B gradient in 60 min (100 min inject-to-inject). Eluting peptides were ionized at a potential of +2 kV and introduced into a Q Exactive mass spectrometer (Thermo Fisher, Bremen, Germany). Intact masses were measured at resolution 70,000 (at *m*/*z* 200) in the Orbitrap using an AGC target value of 3E6 charges. The top 10 peptide signals (charge-states 2+ and higher) were submitted to MS/MS in the higher-energy collision cell (HCD, 1.6 amu isolation width, 25% normalized collision energy). MS/MS spectra were acquired at resolution 17,500 (at *m*/*z* 200) in the orbitrap using an AGC target value of 1E6 charges, a maximum inject time of 60 ms, and an underfill ratio of 0.1%. Dynamic exclusion was applied with a repeat count of 1 and an exclusion time of 30 s.

#### 2.4.3. Protein Identification and Label-Free Quantitation

MS/MS spectra were searched against a Swissprot reference proteome FASTA file (release January 2021; 42,383 entries, including canonical and isoforms, no fragments), using MaxQuant version 2.0.3.0 [18]. Enzyme specificity was set to trypsin, and up to two missed cleavages were allowed. Cysteine carboxyamidomethylation (Cys, +57.021464 Da) was treated as fixed modification, and methionine oxidation (Met, +15.994915 Da) and N-terminal acetylation (N-terminal, +42.010565 Da) as variable modifications. Peptide precursor ions were searched with a maximum mass deviation of 4.5 ppm and fragment ions with a maximum mass deviation of 20 ppm. Peptide and protein identifications were filtered at an FDR of 1% using the decoy database strategy. Proteins that could not be differentiated based on MS/MS spectra alone were grouped into protein groups (default MaxQuant settings). Searches were performed with the label-free quantification option selected. Proteins were quantified by spectral counts (the number of identified MS/MS spectra for the pertinent protein) [19], combining all fractions of a sample. Raw counts were normalized on sample totals, with the normalization factor being determined by the ratio of the average sum across all samples and the sum for a given sample.

#### 2.4.4. Data Processing, Statistics, and Data Deposition

Data were processed and analyzed with in-house R scripts (R version 4.1.3). Heatmaps were generated with modified code from the gplots package. Statistical differences between sample groups were determined with a beta binomial test [20], which takes into account within-sample and between-sample variation. Filtering for differential proteins was performed with a significance cutoff of *p* < 0.05, an absolute fold change cutoff of 1.5, and an additional data abundance cutoff (sum of normalized counts ≥ 5 for at least one condition). The mass spectrometric proteomics data were deposited to the ProteomeXchange Consortium via PRIDE [21] with accession number PXD053910.

### 2.5. siRNA Transfection

When HUVECs reached around 80% confluency, siRNA transfection using Dharmafect reagent 1 (#T-2001, Dharmacon, Lafayette, CO, USA) in OptiMEM (Gibco) was performed. For gene silencing, a final concentration of 25 nM of ON-TARGETplus Human siRNA SMART pool targeting p62, MAGED1, CLASP1, MYO9B, AAMP, MTSS1, or RhoB (all Dharmacon) was used. The ON-TARGET plus non-targeting control pool (N.T.) was used as a negative control. After 7 h, the transfection medium was replaced by normal culture medium. Cells were used for experiments 48 h to 72 h post-transfection.

### 2.6. Lentiviral Production

Lentiviral plasmids for expression of myc-AAMP and Tet3G and control lentiviral plasmids were custom-made and purchased from Vectorbuilder (Chicago, IL, USA). HEK293T cells were transfected with lentiviral packaging and envelope plasmids, pMD2.G (#12259, Addgene, Watertown, MA, USA), pMDLg/pRRE (#12251, Addgene), and pRSV-Rev (#12253, Addgene), and lentiviral plasmids using the transfection reagent PEIpro (#101000017, Polyplus, Illkirch-Graffenstaden, France). Medium was replaced by antibiotic-free culture medium 24 h after transfection. After two days, the lentiviral supernatant was harvested. To remove dead cell debris, the supernatant was spun down and passed through a 0.45 µm PVDF filter. Further, 1 µg/mL of DNase (#79254, Qiagen Venlo, The Netherlands) and 1 mM MgCl_2_ were added to the supernatant and incubated for 20 min at 37 °C. For virus concentration, the supernatant was incubated with a 1/3 volume of Lenti-X Concentrator (#631231, Takara, Kusatsu, Japan) for at least two hours at 4 °C. The lentiviral supernatant was centrifuged at 1500× *g* and 4 °C for 45 min, resuspended in PBS, aliquoted, and stored at −80 °C.

### 2.7. Lentiviral Transduction

HUVECs with 70–80% confluency were used for lentiviral transduction. HUVECs were transduced with control and Tet3G lentivirus or myc-AAMP and Tet3G lentivirus. Successfully transduced cells were selected 24 h after transduction with 1 µg/mL of puromycin (#A1113803, Gibco) for three days. AAMP expression was induced by treatment with 0.1 or 1 µg/mL of doxycycline (#D3447, Merck) 24 h after re-seeding transduced cells. Experiments were conducted 24 h after doxycycline treatment.

### 2.8. Electrical Cell Substrate Impedance Sensing (ECIS)

To measure endothelial barrier function, ECIS was performed [22,23]. HUVECs were seeded on fibronectin-coated ECIS plates (96W10idf) containing gold intercalated electrodes (Applied Biophysics, Troy, NY, USA). Resistance was monitored at 4000 Hz. After siRNA-mediated silencing of genes of interest, the formation of a stable monolayer was assessed.

### 2.9. Western Blot

After siRNA-mediated knockdown or treatment with compounds, as indicated, cells were washed with PBS supplemented with 1 mM CaCl_2_ and 0.5 mM MgCl_2_. Cells were lysed in 2× SDS sample buffer (125 mM Tris-HCl pH 6.8, 4% SDS, 20% glycerol, 100 mM DTT, and 0.02% Bromophenol Blue in MilliQ), and proteins were separated by SDS-PAGE and transferred to a nitrocellulose membrane. Membranes were blocked in 5% BSA in TBS-T for 1 h and incubated with designated primary antibodies in 5% BSA in TBS-T overnight at 4 °C. After incubation with secondary antibodies, proteins were visualized using enhanced chemiluminescence (Amersham/GE Healthcare, Amersham, UK) and an AI600 imager (Amersham/GE Healthcare). Densitometric analysis of the band intensities was performed using ImageQuantTL software (version 8.2.0, Cytiva, Marlborough, MA, USA).

### 2.10. Rac1 and Rho GTPase Activation Assays

Rho activity was analyzed by performing a Rhotekin pulldown, which uses the Rho binding domain of the Rho effector Rhotekin. Confluent HUVECs seeded on fibronectin-coated 55 cm^2^ dishes were washed with ice-cold PBS supplemented with 1 mM CaCl_2_ and 0.5 mM MgCl_2_ and lysed with lysis buffer on ice. RhoA.GTP pulldown was performed using the RhoA Activation Biochem Kit (Cytoskeleton, Denver, CO, USA) according to the manufacturer’s protocol. Input and pulldown samples were analyzed by Western blot.

To analyze Rac1 activity, confluent HUVECs seeded on fibronectin-coated 21 cm^2^ dishes were washed with PBS supplemented with 1 mM CaCl_2_ and 0.5 mM MgCl_2_ and lysed in 500 µL of ice-cold lysis buffer (150 mM NaCl, 50 mM Tris pH 7.6, 1% Triton-X 100, and 20 mM MgCl_2_ in MilliQ) containing 30 μg of biotinylated CRIB peptide. After clearing cell lysates by centrifugation at 14,000 rpm and 4 °C for 5 min, 10% of the supernatant was used as an input sample and mixed with 3× SDS sample buffer (125 mM Tris-HCl pH 6.8, 4% SDS, 20% glycerol, 100 mM DTT, and 0.02% Bromophenol Blue). The remaining lysate was incubated with streptavidin beads (Sigma-Aldrich) for 30 min, rotating at 4 °C. Subsequently, the beads were washed five times with lysis buffer with freshly added 10 mM MgCl_2_. After aspiration of the buffer, the beads were taken up in 30 µL of 2× SDS sample buffer. Input and pulldown samples were analyzed by Western blot.

### 2.11. Scratch Assay and Wound Healing Assay

To measure migration of HUVECs, a scratch assay was performed. A confluent monolayer of HUVECs on a fibronectin-coated culture plate was manually scratched with a pipette tip, and dead cells were removed by washing once with warm medium. Subsequently, the scratch area was imaged every 15 min for 20 h with a Keyence Fluorescence Microscope (BZ-X810, Keyence, Mechelen, Belgium). Closure of the scratch area was analyzed with Image J software (version 1.54d, National Institutes of Health, New York, NY, USA).

For a wound healing assay, HUVECs were seeded on fibronectin-coated ECIS plates (8W10E, Applied Biophysics). When HUVECs reached a stable monolayer, wounding was performed twice at a frequency of 40 kHz with 5 V for 20 s. Migration of HUVECs into the wound was monitored by measuring the resistance at 4000 Hz.

### 2.12. Immunofluorescence Analysis

HUVECs on fibronectin-coated 13 mm coverslips (#0117530, Marienfeld superior, Lauda-Königshofen, Germany) were washed twice with PBS supplemented with 1 mM CaCl_2_ and 0.5 mM MgCl_2_, and subsequently fixed with warm 4% paraformaldehyde in PBS at room temperature for 15 min. Cells were permeabilized with 0.2% Triton X-100 in PBS for 3 min and blocked for 30–60 min with 1% human serum albumin in PBS. Fixed cells were incubated with primary antibodies in 1% human serum albumin overnight at 4 °C. After washing three times with PBS, coverslips were incubated with secondary antibodies, Acti-stain 670 phalloidin (#PHDN1-A, Cytoskeleton) or DAPI (#62248, Thermo Fisher Scientific), for 1 h at room temperature. Subsequently, coverslips were mounted with Mowiol4-88/DABCO solution (Calbiochem, Sigma-Aldrich, St. Louis, MO, USA). Confocal scanning laser microscopy was performed on a Nikon A1R confocal microscope (Nikon, Tokyo, Japan). Images were analyzed and equally adjusted with ImageJ (version 1.54d, National Institutes of Health, USA) software. In order to measure the cell area, binary images were created, and the area of the cell was measured by the analyze tool in Image J.

### 2.13. Statistical Analysis

Experimental data are presented as mean + SD or ±SD. Statistical analysis was performed using GraphPad Prism (version 10, GraphPad Software, Boston, MA, USA). One-way ANOVA with Dunnett’s post hoc test was applied when several groups were compared to one (control) group. When only two groups were compared, the two-tailed Student’s *t*-test was used. *p*-values < 0.05 were considered as statistically significant.

## 3. Results

### 3.1. Differentially Expressed Proteins Identified upon Inhibition of Ubiquitination

To identify novel regulators of endothelial barrier function, which have a short half-life and are tightly regulated by ubiquitination, a proteomics screen was performed. To inhibit ubiquitination, HUVECs were treated with the E1 ligase inhibitor MLN7243, the Cullin-based E3 ligase inhibitor MLN4924, or DMSO as a control. Treatment was performed for 6 h only, to limit positive hits to those proteins that have a high turnover (i.e., short half-life) and for which ubiquitination is potentially critical to control their function and downstream signaling. For further description of preparation of proteomics samples, see the Methods Section. A Coomassie staining was used to ensure equal protein amounts between lysates (Appendix A).

In total, 3974 proteins were detected in the proteomics screen (Figure 1A,B). To identify differentially expressed proteins, two comparisons were performed: proteins found in MLN7243- compared to DMSO-treated HUVECs and proteins found in MLN4924- compared to DMSO-treated HUVECs. Supervised clustering for both comparisons showed separate clusters for DMSO- and inhibitor-treated samples, indicating differential protein expression profiles (Appendix A). For both comparisons, a significance cutoff (*p* < 0.05) was applied (Figure 1A,C). Interestingly, the majority of the 90, statistically significant, differentially expressed proteins between MLN7243- and DMSO-treated ECs showed a positive fold change (Figure 1C). This means they were upregulated within the short timeframe of the experiment, indicating that turnover of the respective proteins was fast and controlled by ubiquitination. Intriguingly, the Rho GTPase RhoB, whose stability was previously shown to be tightly regulated by ubiquitination in ECs [16,17], was significantly upregulated in MLN7243 lysates (Figure 1C,D; Table 1). Similarly, the majority of 85 significantly differentially expressed proteins between MLN4924- and DMSO-treated HUVECs showed a positive fold change (Figure 1E), suggesting that Cullin-based E3 ligases regulate ubiquitination and stability of the respective hits.

For further data analysis, a data abundance filter (sum of normalized count ≥ 5 for at least one condition) and fold change cutoffs (±1.5) were applied for each comparison, resulting in 28 upregulated (Table 1) and 4 downregulated proteins (Table 2) in HUVECs treated with MLN7243 and 26 upregulated (Table 3) and 6 downregulated proteins (Table 4) in MLN4924-treated HUVECs (Figure 1A,B). According to UniProt, these proteins are associated with various cellular functions, such as cell death, ubiquitination, transcription, metabolism, and the cytoskeleton (Table 1, Table 2, Table 3 and Table 4). A comparison of differentially expressed proteins in MLN7243- and MLN4924-treated lysates after data analysis revealed four overlapping proteins (Appendix A): CLASP1, SQSTM1, RPRD1B, and HMOX1. This indicated that, according to the proteomics screen, E1 ligase and Cullin-based E3 ligases regulated the ubiquitination and turnover of different sets of proteins in ECs, within the timeframe of the experiment. In total, sixty significantly dysregulated proteins were identified in EC lysates upon inhibition of E1 ligase or Cullin-based E3 ligases, suggesting that ubiquitination determined turnover of a specific set of proteins in ECs, potentially regulating their protein function and signaling ability.

To further examine which proteins, identified in the proteomics screen, could play a role in regulating endothelial barrier function, a literature search was performed. Based on their previously described role in regulation of the cytoskeleton or Rho GTPase signaling, EC function, or regulation of their function by ubiquitination, seven proteins were identified as potentially promising hits. Figure 1D,F show the normalized peptide counts in the proteomics lysates of these proteins, which include MAGED1, SQSTM1 (which is also known as p62), ArhGAP24, CLASP1, MYO9B, AAMP, and MTSS1.

### 3.2. AAMP and MTSS1 Regulate Endothelial Barrier Function and Have a Short Half-Life

To further dissect the potential role of MAGED1, SQSTM1/p62, CLASP1, MYO9B, AAMP, and MTSS1 in regulating endothelial barrier function, siRNA-mediated silencing of these promising hits was performed, and endothelial barrier function was analyzed in real time using electric cell-substrate impedance sensing (ECIS). ArhGAP24 was excluded from further analysis, as we recently reported that depletion of ArhGAP24 did not affect endothelial integrity [22]. Since RhoB is a known negative regulator of endothelial integrity with a short half-life of 2–3 h [8,24], knockdown of RhoB was included as a positive control. Then, 48 h after siRNA transfection, depletion of AAMP resulted in a significant increase in trans-endothelial electrical resistance (Figure 2A and Appendix A), reflecting an improved barrier function. In several, but not all, experiments, the increased barrier function induced by the loss of AAMP declined after >50 h. This could indicate that AAMP plays a relatively prominent role during EC adhesion, spreading, and barrier formation, as compared to barrier maintenance for prolonged timeframes. The results could also be explained by the limited duration of the siRNA effects in some of the experiments. Similarly, knockdown of MTSS1 improved resistance 72 h after siRNA transfection (Figure 2A and Appendix A). Accordingly, ECIS measurements over time demonstrated a steady increase in barrier function of HUVECs depleted of AAMP or MTSS1 compared to the N.T. control (Figure 2B). Concomitantly, the successful depletion of AAMP, MTSS1, CLASP1, and SQSTM1/p62 was confirmed (Figure 2C,D and Appendix A). Thus, these data identify AAMP and MTSS1 as novel negative regulators of endothelial barrier function.

To confirm upregulation of AAMP and MTSS1 after blocking ubiquitination in a different set of protein lysates, HUVECs were treated for 2 h, 4 h, and 6 h with MLN4924. Notably, MTSS1 was upregulated up to 3.4-fold, coinciding with accumulation of the positive control RhoB (Appendix A). AAMP showed a slight upregulation of 1.3-fold after treatment with MLN4924 (Appendix A). In line with that notion, Western blot analysis of lysates used for the proteomics screen confirmed a 2-fold increase of AAMP, as well as a 7-fold accumulation of RhoB, following MLN4924 treatment (Appendix A). These data indicated that the protein abundance of AAMP and MTSS1, similar to RhoB, was controlled by Cullin-based E3 ligases.

Upregulation of AAMP and MTSS1 after 6 h of MLN4924 treatment suggests a short half-life of both proteins. In agreement with this, blocking protein translation with cycloheximide (CHX) confirmed that AAMP and RhoB share a similar half-life (~2–4 h, Figure 2E), whereas the data suggested that MTSS1 has an even shorter half-life than RhoB of about 1 h (Figure 2F). Taken together, these data demonstrate that AAMP and MTSS1 are negative regulators of endothelial integrity, of which their protein turnover is tightly regulated by ubiquitination.

### 3.3. AAMP Regulates Endothelial Barrier Function by Controlling RhoA and RhoB Activity

Since depletion of AAMP showed the most significant effect on endothelial integrity, the role of AAMP in endothelial (barrier) function was further examined. For the following experiments, knockdown of AAMP was performed with an siRNA pool and a single siRNA, both targeting AAMP. Similar to the previously used siAAMP pool, siAAMP #1-mediated silencing of AAMP led to a significant increase in trans-endothelial electrical resistance (Figure 3A and Appendix A), while inducing a reduction of AAMP expression similar to the siAAMP pool (Figure 3B). First, the effect of AAMP depletion on VE-cadherin levels was assessed. Importantly, VE-cadherin levels remained unaltered (Appendix A), indicating that differences in VE-cadherin stability or expression were not responsible for the improvement of the endothelial barrier with siAAMP. To further assess the role of AAMP in EC function, siRNA-mediated depletion of AAMP was performed and cell size was measured. Silencing of AAMP led to a significant increase in cell area compared to the N.T. control (Figure 3C), indicative of cell spreading. However, Rac1 activity, typically mediating cell spreading, remained mostly unaffected by AAMP depletion (Appendix A). Accordingly, total protein levels of Rac1 and Cdc42 remained unaltered (Appendix A).

Next, it was hypothesized that the increase in cell size was not mediated by elevated cell spreading but impaired cell contraction. Consequently, the level of RhoA activity following AAMP depletion was assessed. Interestingly, the data showed a reduction in active RhoA, compared to N.T. (Figure 3D). Strikingly, not only RhoA activity was affected but also total RhoA protein levels declined slightly, but significantly (Figure 3E), suggesting that AAMP not only regulates RhoA activity levels but also RhoA stability. Since RhoA and RhoB share similar cellular functions, it was examined if AAMP affects RhoB activity in a similar manner. Indeed, silencing of AAMP caused a significant reduction in RhoB.GTP (Figure 3F) and a minor decrease in RhoB total levels for the siAAMP pool, but not for siAAMP #1 (Figure 3G). These data suggest that AAMP controls RhoA and RhoB turnover, which affects RhoA and RhoB activity levels.

According to the literature, both RhoA.GTP and RhoB.GTP induce phosphorylation of myosin light chain (pMLC), leading to actomyosin-based contractility. Unexpectedly, an immunofluorescent staining revealed that pMLC increased upon depletion of AAMP (Figure 3H). Since protein levels of pMLC remained unaltered (Figure 3I), the increase in pMLC may be due to local changes, redistribution of the cellular pool of pMLC, or even a change in the levels of mono- vs. di-phosphorylation of MLC. Notably, silencing of AAMP did not result in changes in F-actin or VE-cadherin distribution (Figure 3H). Taken together, siRNA-mediated loss of AAMP reduced the activity of RhoA and RhoB, which potentially diminished the ability of RhoA and RhoB to induce cellular contraction in resting ECs, disturbing the balance of spreading and contraction. The decrease in contraction at rest could potentially explain the observed increase in cell spreading and endothelial barrier function.

### 3.4. AAMP Potentially Regulates EC Migration

Previously, it has been shown that AAMP plays a role in cell migration in various cancer cell types, but also in vascular smooth muscle cells and ECs [25,26,27,28]. Consistent with previous findings, depletion of AAMP led to a delayed wound closure compared to N.T. control in a wound healing assay performed with ECIS (Figure 4A). Conversely, silencing of AAMP did not affect the closure of a scratch in a HUVEC monolayer (Appendix A). Taken together, these data indicate that AAMP may play a positive regulatory role in regulating EC migration.

### 3.5. AAMP Colocalizes with F-Actin and Cortactin at Membrane Ruffles

To further decipher the role of AAMP in EC function, an AAMP overexpression system was used, which enabled doxycycline-inducible expression of AAMP (Figure 5A). Interestingly, AAMP overexpression did not induce an effect on trans-endothelial electrical resistance (Appendix A), suggesting that an excess of AAMP does not affect endothelial integrity, in contrast to the absence of AAMP (Figure 2A,B). In agreement with this, AAMP overexpression did not regulate total RhoA or RhoB protein levels (Appendix A).

Since immunostaining for endogenous AAMP was not informative, we used ectopically expressed AAMP to study its localization. Immunofluorescent staining of a confluent HUVEC monolayer revealed AAMP expression in the nucleus and cytosol; however, AAMP was not localized at the plasma membrane, here identified by VE-cadherin (Figure 5B). Interestingly, in a sub-confluent monolayer of HUVECs, AAMP was colocalizing with F-actin at membrane ruffles; however, not with stress fibers, indicating specific targeting of AAMP to cortical actin in protrusions (Figure 5C). Membrane ruffles can be found at the leading edge of a migrating cell and require dynamic remodeling of actin filaments. Formation of actin filament branches from already existing filaments is initiated by the Arp2/3 complex, a process which is crucial for actin network formation in membrane ruffles [29]. Previously, the WD40 repeat domain-containing protein Coronin1B was found to induce Arp2/3 dissociation, thereby causing actin filament disassembly and antagonizing cortactin-mediated Arp2/3 stabilization [30]. Since AAMP also contains a WD40 repeat domain, based on 8 WD40 repeats, similar to Coronin1B [31], and localizes to membrane ruffles, it was investigated if AAMP plays a role in controlling actin branching.

First, immunofluorescent staining of AAMP and cortactin of a sub-confluent monolayer of HUVECs was performed. Interestingly, AAMP and cortactin colocalized at membrane ruffles (Figure 5D). A myc-co-IP was performed in order to assess if AAMP binds to cortactin. However, no direct binding of AAMP to cortactin was observed (Figure 5E) unlike coronin1B and cortactin [29]. Differential colocalization of AAMP with p34-Arc, a member of the Arp2/3 complex, remains unclear. AAMP was enriched at membrane ruffles; however, free GFP localized similarly to most membrane ruffles, indicating a volume effect rather than targeted localization of AAMP to membrane ruffles (Appendix A). AAMP was not found to bind directly to p34-Arc (Appendix A). Notably, overexpression of AAMP did not lead to improved wound closure in a scratch assay and ECIS wounding assay (Appendix A), suggesting that AAMP is necessary but not sufficient to promote EC migration. Taken together, these findings indicate that AAMP potentially localizes to membrane ruffles; however, the role of AAMP in regulation of these F-actin-rich structures remains to be elucidated in more detail.

## 4. Discussion

In this study, we described the critical role of protein turnover for EC, particularly for endothelial barrier function. Using a proteomics approach to define the short-term effects of inhibition of E1 ligases or Cullin-mediated E3 ligases, we identified fifty significantly upregulated and ten significantly downregulated proteins. These data indicated that ubiquitination mainly served to limit protein stability. However, whether these proteins are directly ubiquitinated and degraded, or regulated indirectly via another pathway, remains unknown. Since the E1 ligases act upstream of the Cullin-mediated E3 ligases, it was expected that dysregulated proteins determined in the MLN7243–DMSO comparison would include all proteins from the MLN4924–DMSO comparison. However, according to the results of the screen, only four proteins overlapped, namely, CLASP1, SQSTM1, RPRD1B, and HMOX1 (Appendix A). These results could be explained by the fact that the proteomics analysis was limited to identify around 4000 proteins, hence, the mode of detection was not sufficiently sensitive to detect absolutely all proteins, and a considerable number remained under the detection limit. Moreover, there are many types of E3 ligases, so while E1 inhibition will affect all ubiquitination substrates in a cell, inhibition of Cullin E3 ligases will only affect a subset of these.

The identified proteins are associated with a range of cellular processes, indicating that protein regulation by ubiquitination and degradation is important for various EC functions. Intriguingly, a considerable number of dysregulated proteins identified following E1 inhibition were chaperones of misfolded and unfolded proteins and linked to ubiquitination and cell death (Table 1 and Table 2). This suggests induction of an unfolded protein response as a consequence of the inhibition of E1 ligase-mediated degradation, in line with what was previously shown [30,31]. Significantly up- or down-regulated proteins following inhibition of Cullin E3 ligases showed a considerable variety of cellular processes compared to E1 ligase inhibition, indicating specific accumulation or reduction of proteins rather than an overall cellular response to the blocking of ubiquitination. Notably, a considerable number of proteins related to actin cytoskeleton dynamics were identified. However, the precise implications of protein turnover of these hits regarding their signaling capacity remains to be elucidated.

Subsequently, we sought to identify which proteomics hits regulated endothelial integrity by performing a small siRNA screen with potentially promising proteomics hits, identified by a literature search focused on the previously described role in regulation of cytoskeleton dynamics, Rho GTPases, or EC function. AAMP and MTSS1 were identified, for the first time, to negatively regulate endothelial barrier function. In addition, both proteins showed a short half-life, similar to RhoB, in line with the proteomics analysis (Figure 6).

MTSS1, also called Missing in Metastasis (MIM), was originally identified in human bladder cancer cell lines. Lee et al. reported decreased MTSS1 levels in metastatic bladder and prostatic cancer cell lines [32]. MTSS1 contains the actin-monomer binding motif WH2 and has been shown to promote F-actin formation via Arp2/3 complex [33,34,35]. Interestingly, MTSS1 was described to localize at junctions in kidney epithelium and squamous carcinoma cells through an I-BAR domain, and to be essential for maintenance of cell–cell adhesion by promoting Rac1 activation [34,36].

Here, we show that MTSS1 has a short half-life and is a novel regulator of endothelial barrier function. In contrast to earlier studies [34,36], depletion of MTSS1 leads to improved barrier function, indicating a negative effect of MTSS1 on endothelial integrity. However, Lin et al. described a dual function of MTSS1 on actin polymerization [35], suggesting different predominant functions of MTSS1 in epithelial and endothelial cells. The molecular mechanisms underlying MTSS1-mediated regulation of the endothelial barrier remain a subject for further research.

AAMP was originally identified in a human melanoma cell line in a search for proteins crucial to migration but is additionally expressed in various other cell types [37]. AAMP is highly expressed in different types of cancer, such as gastrointestinal stromal tumors and ductal carcinoma in situ of the breast with necrosis [38,39]. Increased AAMP levels are linked to poor prognosis, increased cancer cell migration, and invasion in breast cancer, human non-small cell lung cancer, and colorectal cancer [25,26,40]. A recent study reported that AAMP has high potential as a prognostic, pan-cancer marker [41]. Interestingly, AAMP was found to be strongly expressed in ECs in various human tissues, and in different cell types of the vasculature [37,42]. Further, AAMP was shown to drive angiogenesis and progression of atherosclerosis and restenosis [27,28,43]. Thus, we aimed to further understand the role of AAMP in EC, focusing on its role in endothelial barrier function.

Functionally, silencing of AAMP resulted in increased cell spreading and a decrease in RhoA and RhoB activity, indicative of impaired contraction, and potentially responsible for the improvement in barrier function. Interestingly, depletion of AAMP also led to a small but significant reduction in the RhoA total protein level, and a slight decrease in the RhoB protein level, both suggesting that AAMP controls stability of RhoA as well as RhoB to a limited extent, which might be the underlying cause for reduced GTP-bound active forms of RhoA/B (Figure 6).

Our data provide the first insights that AAMP regulates RhoB stability and activity. Regulation of RhoA by AAMP has been previously shown in colon cancer cells, vascular smooth muscle cells, and ECs [25,27,28]. However, previously, the effect of AAMP depletion on RhoA activity in ECs has only been shown indirectly by analyzing the membrane-associated fraction of RhoA [28]. Our data thus confirmed that AAMP positively regulated the actual binding of GTP to both RhoA and RhoB and was required for their basal activity in human endothelium. Regulation of total RhoA protein levels by AAMP has previously only been described in colon cancer cells, not ECs [25].

Mechanistically, we did not observe a consistent downstream effect of AAMP silencing and reduced RhoA/B activity on F-actin distribution in quiescent ECs. Unexpectedly, an increased pMLC staining was observed upon silencing of AAMP. Since total pMLC was not increased and no stress fiber formation was observed, pMLC might be only elevated on a local level, presumably even promoting adhesion of cell–cell contacts, as previously reported [44,45].

Notably, overexpression of AAMP, in turn, did not lead to improvement in endothelial barrier function, accumulation of RhoA or RhoB, or an increase in F-actin stress fibers’ formation, indicating that it is the lack of AAMP that interferes with EC function, while an excess of AAMP does not induce EC dysfunction. This could be due to downstream signaling or targeting proteins, of which the concentration is limiting, so that any effects of the ectopically expressed AAMP are not detected.

When AAMP was identified by Beckner et al., it was suggested that AAMP has a membrane-associated domain [37]. In agreement with this, major localization of AAMP to the plasma membrane was confirmed in vascular smooth muscle cells by cellular fractionation, and a potential transmembrane domain was predicted [27]. Cellular fractionation and immunofluorescent staining of an EC line, of human aortic smooth muscle cells, and of HEK cells revealed that AAMP localizes to the membrane as well as to the soluble fraction [42]. Intriguingly, in our AAMP overexpression studies, AAMP localization was limited largely to the cytosol, and AAMP did not colocalize with VE-cadherin at junctions. This finding is in agreement with the previously described predominant cytosolic distribution of AAMP in HeLa cells and HUVECs [28,46]. This suggests that AAMP localization shows marked differences between cell types, further suggesting a cell-type-specific role for AAMP.

Colocalization of AAMP with F-actin, cortactin, and Arp2/3 in membrane ruffles further suggests a role of AAMP in cell migration and proposes a role of AAMP in regulation of F-actin (Figure 6). However, these data have to be interpreted with caution, since AAMP was colocalizing in part with the volume marker GFP at membrane ruffles. AAMP is a WD40 repeat-containing protein [37], which typically acts as a scaffold in multimeric protein complexes [47]. Previously, it was described that the WD40 domain-containing protein coronin 1B localizes at membrane ruffles and coordinates actin branching by cortactin and the Arp2/3 complex controlling protrusions and cell migration [29,48]. Notably, coronin 1B is recruited to endothelial cell–cell junctions and contributes to actin organization in ECs [49], suggesting a similar role for AAMP in the regulation of actin dynamics and endothelial junctions. To confirm localization of AAMP at membrane ruffles and if AAMP directly regulates F-actin branching via the Arp2/3 complex, further investigations are required.

AAMP is ubiquitously expressed and has various cellular functions. Its subcellular localization and role in various pathways and diseases have been reported; hence, AAMP is a multifunctional protein. Previously, the role of AAMP in angiogenesis and migration has been described in ECs [27,50,51]. The current study reported on the critical role and downstream mechanisms of AAMP in endothelial barrier function, broadening the understanding of AAMP signaling in EC function.

This study provided insights on the crucial role of fast protein turnover in quiescent endothelial barrier function and identified two proteins with a short half-life, AAMP and MTSS1, as negative regulators of endothelial integrity. These data thus contribute to unraveling the mechanisms that control endothelial integrity and extend our knowledge on (patho)physiological vascular leakage.

## Figures and Tables

**Figure 1 cells-13-01609-f001:**
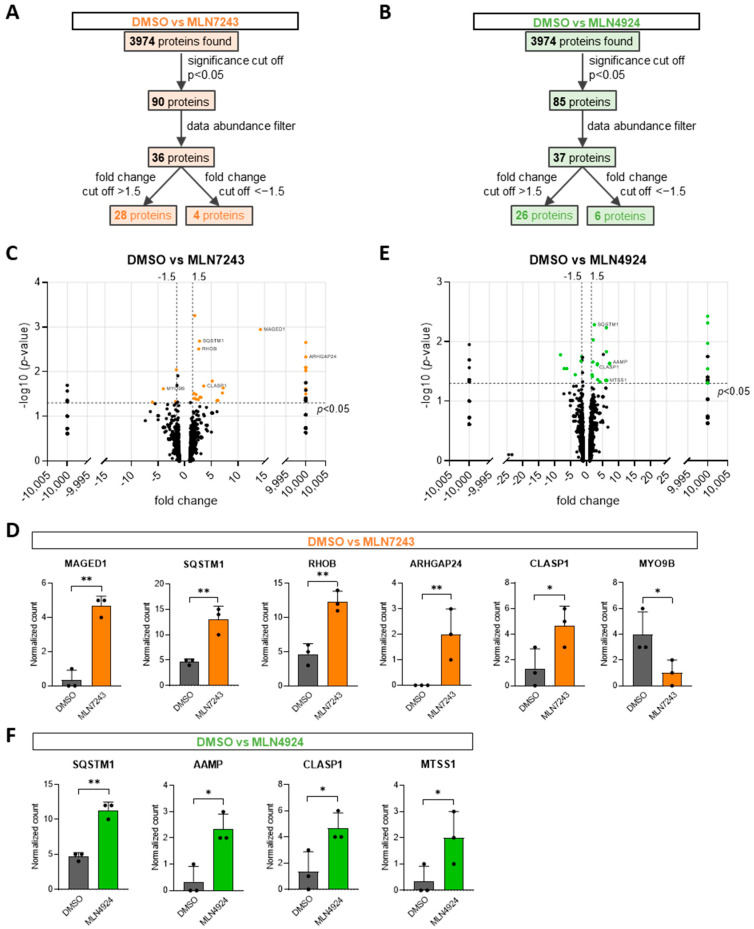
Proteomics screen to identify up- or down-regulated proteins upon inhibition of ubiquitination. (**A**,**B**) Flow chart on the analysis of the identified proteins in proteomics, for (**A**) DMSO vs. MLN7243 comparison and (**B**) DMSO vs. MLN4924 comparison. (**C**,**E**) Volcano plots of all identified proteins for each comparison. Significance cutoff and fold change cutoffs are marked with dashed lines. Orange marked proteins represent the 32 proteins after data analysis from (**A**), while green marked proteins represent the 32 proteins after data analysis from (**B**). (**D**,**F**) Normalized peptide counts of proteomics data for proteins after data analysis from (**A**,**B**), which are potentially involved in endothelial barrier regulation, (**D**) for DMSO vs. MLN7243 comparison and (**F**) for DMSO vs. MLN4924 comparison. * *p* < 0.05; ** *p* < 0.01.

**Figure 2 cells-13-01609-f002:**
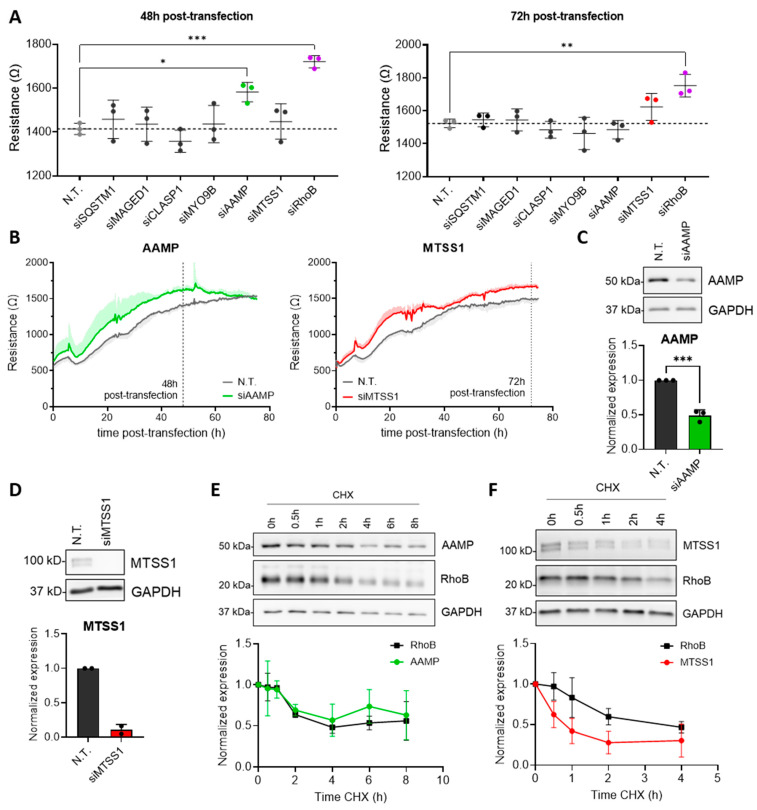
AAMP and MTSS1 are novel negative regulators of endothelial barrier function. (**A**) HUVECs were transfected with N.T., siSQSTM1, siMAGED1, siCLASP1, siMYO9B, siAAMP, siMTSS1, or siRhoB. Resistance of HUVECs 48 h and 72 h post-transfection is shown. The dotted lines show the respective N.T. values. Data are presented as mean ± SD, n = 3. (**B**) HUVECs were transfected with N.T., siAAMP, or siMTSS1, and resistance was measured over time. The graphs represent the time course of one representative experiment, depicted in (**A**), as mean + SD. (**C**) HUVECs were transfected with N.T. or siAAMP. Western blot analysis of AAMP expression 48 h post-transfection. Bar graph shows quantified AAMP expression normalized to GAPDH and N.T. Data are presented as mean + SD, n = 3. (**D**) HUVECs were transfected with N.T. or siMTSS1. Western blot analysis of MTSS1 expression 72 h post-transfection. Bar graph shows quantified MTSS1 expression normalized to GAPDH and N.T. Data are presented as mean + SD, n = 2. (**E**,**F**) Western blot analysis of (**E**) AAMP and RhoB and (**F**) MTSS1 and RhoB expression after 25 µg/mL CHX treatment for indicated time points. Graphs show quantification of (**E**) AAMP and RhoB and (**F**) MTSS1 and RhoB expression normalized to respective GAPDH and 0 h CHX. Data are presented as mean ± SD, n = 3–4. * *p* < 0.05, ** *p* < 0.01, and *** *p* < 0.001. HUVEC, human umbilical vein endothelial cell; N.T., non-targeting siRNA; CHX, cycloheximide.

**Figure 3 cells-13-01609-f003:**
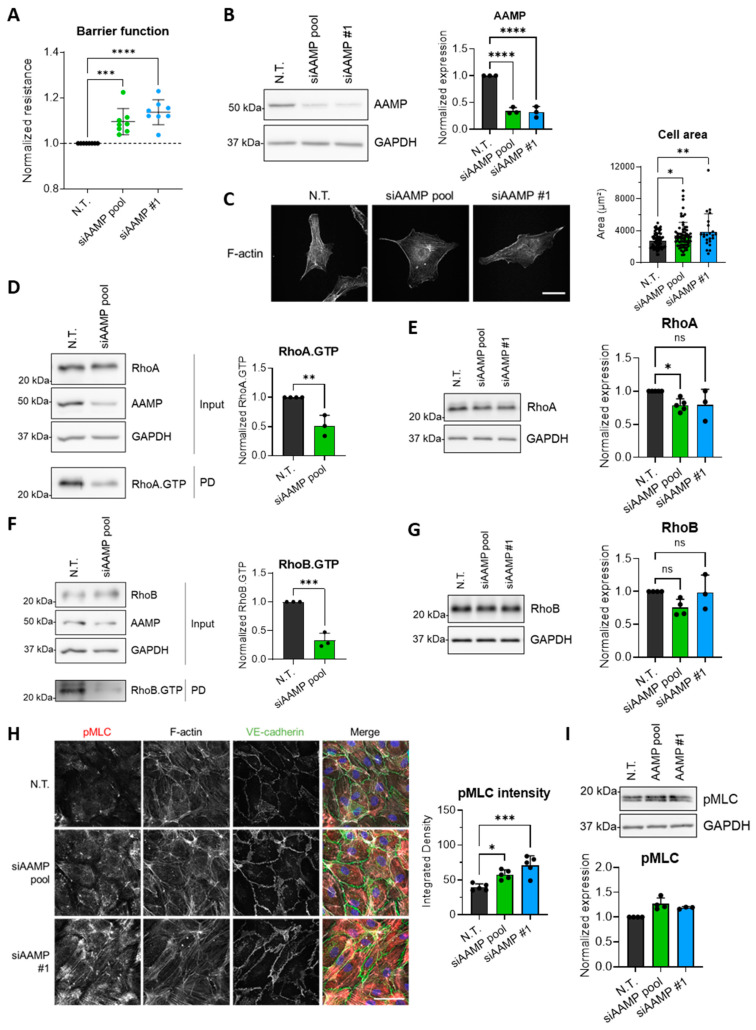
Silencing of AAMP increases cell size and affects RhoA/B expression and activity. (**A**–**I**) HUVECs were transfected with N.T., siAAMP pool, or siAAMP #1. (**A**) Normalized resistance of HUVECs 48 h post-transfection. Data are presented as mean ± SD, n = 8. (**B**) Western blot analysis for AAMP expression. Bar graph shows quantification of AAMP expression normalized to GAPDH and N.T. Data are presented as mean + SD, n = 3. (**C**) Immunofluorescent staining for F-actin (white) of single HUVECs. Scale bar represents 50 µm. Bar graph shows cell area of 23–67 cells per condition. (**D**) Rhotekin pulldown followed by Western blot analysis for the GTP-bound, active form of RhoA. Bar graph shows quantification of RhoA.GTP normalized to GAPDH and N.T. Data are presented as mean + SD, n = 3. (**E**) Western blot analysis for RhoA expression. Bar graph shows quantification of RhoA expression normalized to GAPDH and N.T. Data are presented as mean + SD, n = 3–5. (**F**) Rhotekin pulldown followed by Western blot analysis for the GTP-bound, active form of RhoB. Bar graph shows quantification of RhoB.GTP normalized to GAPDH and N.T. Data are presented as mean + SD, n = 3. (**G**) Western blot analysis for RhoB expression. Bar graph shows quantification of RhoB expression normalized to GAPDH and N.T. Data are presented as mean + SD, n = 3–4. (**H**) Immunofluorescent staining for pMLC (red), F-actin (white), and VE-cadherin (green), and counterstained with DAPI (blue). Scale bar represents 50 µm. Bar graph shows fluorescent intensity of pMLC staining. Data are presented as mean + SD. (**I**) Western blot analysis for pMLC expression. Bar graph shows quantification of pMLC expression normalized to GAPDH and N.T. Data are presented as mean + SD, n = 3–4. * *p* < 0.05, ** *p* < 0.01, *** *p* < 0.001, and **** *p* < 0.0001. HUVEC, human umbilical vein endothelial cell; N.T., non-targeting siRNA; PD, pulldown; pMLC, phosphorylated myosin light chain; ns, non-significant.

**Figure 4 cells-13-01609-f004:**
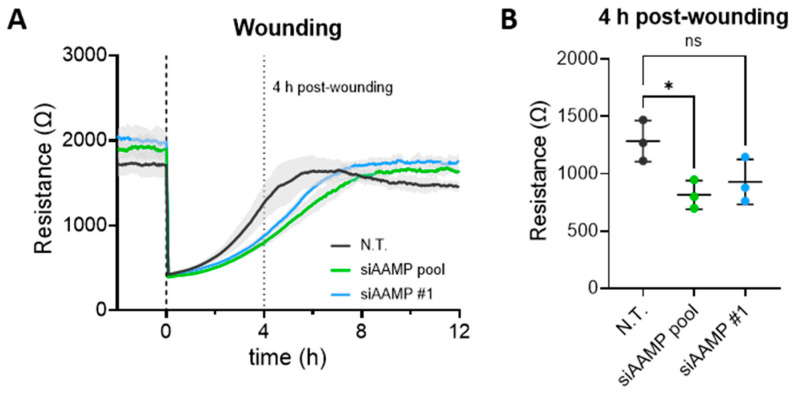
Silencing of AAMP impairs EC migration. (**A**,**B**) After transfection of HUVEC with N.T., siAAMP pool, or siAAMP #1 (1–2 wells per condition), the monolayer was wounded by a high electrical current. Re-migration of HUVECs into the wound was monitored by resistance measurement with ECIS. (**A**) One representative measurement. (**B**) Scatter plot shows absolute resistance values 4 h after wounding. Data presented are the summary of three independent experiments and presented as mean ± SD, n = 3. * *p* < 0.05. HUVEC, human umbilical vein endothelial cell; N.T., non-targeting siRNA; ns, non-significant.

**Figure 5 cells-13-01609-f005:**
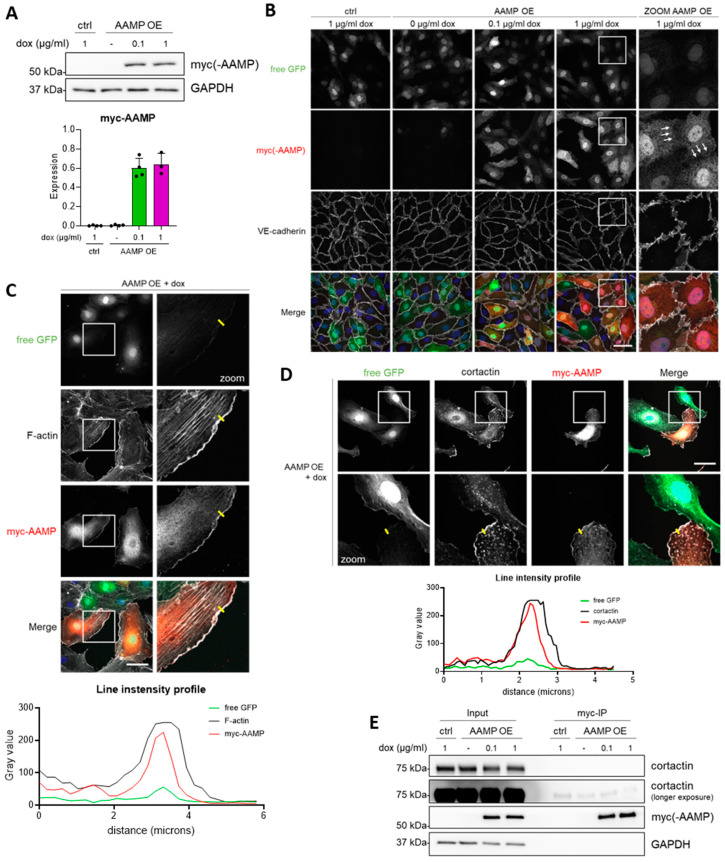
AAMP colocalizes with F-actin and cortactin in membrane ruffles. (**A**–**E**) HUVECs were transduced with ctrl or AAMP lentivirus and treated with the indicated concentrations of doxycycline for 24 h. (**A**) Western blot analysis for myc-AAMP expression. Bar graph shows quantification of myc-AAMP expression normalized to GAPDH. Data are presented as mean + SD, n = 3–4. (**B**) Immunofluorescent staining for GFP (green), myc-AAMP (red), and VE-cadherin (white), and counterstained with DAPI (blue). Scale bar represents 50 µm. (**C**) Immunofluorescent staining of sub-confluent HUVECs for GFP (green), F-actin (white), and myc-AAMP (red), and counterstained with DAPI (blue). Scale bar represents 50 µm. Graph shows gray values of all channels along the yellow line drawn in zoom images. (**D**) Immunofluorescent staining of sub-confluent HUVECs for GFP (green), cortactin (white), and myc-AAMP (red), and counterstained with DAPI (blue). Scale bar represents 50 µm. Graph shows gray values of all channels along the yellow line drawn in zoom images. (**E**) Myc-co-IP followed by Western blot analysis for myc-AAMP and cortactin. HUVEC, human umbilical vein endothelial cell; ctrl, control; dox, doxycycline; OE, overexpression.

**Figure 6 cells-13-01609-f006:**
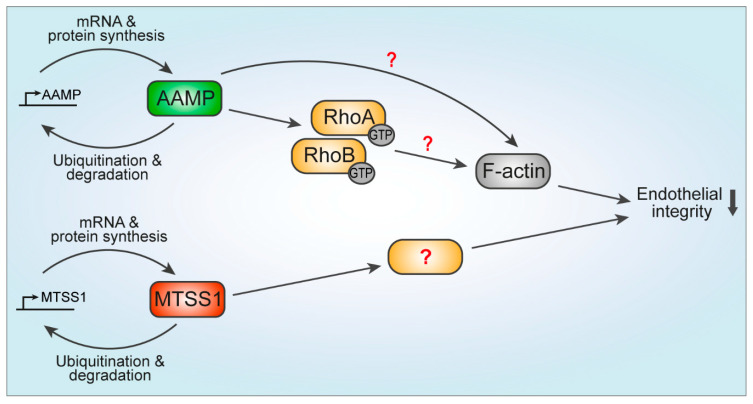
Model of AAMP and MTSS1 regulating endothelial integrity. AAMP and MTSS1 have a short half-life, hence, their turnover is tightly regulated by continuous mRNA expression, protein synthesis, and ubiquitin-mediated degradation. Mechanistically, AAMP controls stability and promotes activation of RhoA and RhoB, leading to impaired endothelial barrier function. Potentially, AAMP affects F-actin dynamics directly. MTSS1 is a novel negative regulator of endothelial integrity; however, the mechanism of regulation remains to be elucidated.

**Table 1 cells-13-01609-t001:** The 28 significantly upregulated proteins identified in lysates of MLN7243-treated ECs after data analysis, as depicted in Figure 1A. Proteins are ordered by increasing *p*-value. Fold change 10,000 denotes uniquely expressed proteins. * According to UniProt.

Protein Abbreviation	Protein Name	UniProt ID	Associated with *	*p*-Value	Fold Change
HSPA1A; HSPA1B	Heat shock 70 kDa protein 1A; Heat shock 70 kDa protein 1B	P0DMV8; P0DMV9	molecular chaperone	0.00055	1.88
MAGED1	Melanoma-associated antigen D1	Q9Y5V3	ubiquitination, cell cyle	0.0011	14.24
SQSTM1	Sequestosome-1	Q13501	autophagy	0.0021	2.81
CDKN1A	Cyclin-dependent kinase inhibitor 1	P38936	cell cycle	0.0022	10,000
RHOB	Rho-related GTP-binding protein RhoB	P62745	apoptosis, cell adhesion, protein trafficking	0.0031	2.64
ARHGAP24	Rho GTPase-activating protein 24	Q8N264	cytoskeletal organization	0.0047	10,000
DAPK3	Death-associated protein kinase 3	O43293	autophagy, cytsoekeletal organization	0.0081	10,000
HK2	Hexokinase-2	P52789	glycolysis	0.0081	10,000
RPRD1B	Regulation of nuclear pre-mRNA domain-containing protein 1B	Q9NQG5	transcription	0.0081	10,000
PJA2	E3 ubiquitin-protein ligase Praja-2	O43164	ubiquitination	0.0081	10,000
ID3	DNA-binding protein inhibitor ID-3	Q02535	transcription	0.0081	10,000
HERPUD1	Homocysteine-responsive endoplasmic reticulum-resident ubiquitin-like domain member 1 protein	Q15011	degradation of misfolded proteins	0.0095	10,000
TNFRSF10B	Tumor necrosis factor receptor superfamily member 10B	O14763	apoptosis	0.016	5.18
CLASP1	CLIP-associating protein 1	Q7Z460	microtubules	0.021	3.59
MORF4L1	Mortality factor 4-like protein 1	Q9UBU8	transcription	0.023	7.24
ANKRD1	Ankyrin repeat domain-containing protein 1	Q15327	transcription? cell death?	0.024	10,000
ATXN3	Ataxin-3	P54252	deubiquitin-ation	0.030	7.10
DNAJA1	DnaJ homolog subfamily A member 1	P31689	chaperone	0.031	1.74
LDLR	Low-density lipoprotein receptor	P01130	cholesterol metabolism	0.032	10,000
TXLNA	Alpha-taxilin	P40222	vesicle trafficking?	0.033	2.15
BAG3	BAG family molecular chaperone regulator 3	O95817	molecular chaperone	0.037	2.87
HMGCS1	Hydroxymethylglutaryl-CoA synthase, cytoplasmic	Q01581	cholesterol synthesis	0.038	3.08
HMOX1	Heme oxygenase 1	P09601	degradation of heme	0.040	1.77
NDUFS1	NADH-ubiquinone oxidoreductase 75 kDa subunit, mitochondrial	P28331	oxidative phosphory-lation	0.041	2.15
PAFAH1B2	Platelet-activating factor acetylhydrolase IB subunit beta	P68402	hydrolysis of platelet-activating factor	0.042	2.55
SPAG7	Sperm-associated antigen 7	O75391	acetylation?	0.044	6.22
KCMF1	E3 ubiquitin-protein ligase KCMF1	Q9P0J7	ubiquitination	0.045	6.20
PBDC1	Protein PBDC1	Q9BVG4	?	0.045	6.05

**Table 2 cells-13-01609-t002:** The 4 significantly downregulated proteins identified in lysates of MLN7243-treated ECs after data analysis, as depicted in Figure 1A. Proteins are ordered by increasing *p*-value. * According to UniProt.

Protein Abbreviation	Protein Name	UniProt ID	Associated with *	*p*-Value	Fold Change
HNRNPM	Heterogeneous nuclear ribonucleoprotein M	P52272	mRNA splicing and processing	0.0091	−1.55
MYO9B	Unconventional myosin-IXb	Q13459	motor protein, GAP	0.024	−4.00
KRT2	Keratin, type II cytoskeletal 2 epidermal	CON__P35908v2		0.047	−1.71
KRT16	Keratin, type I cytoskeletal 16	CON__P08779		0.048	−6.05

**Table 3 cells-13-01609-t003:** The 26 significantly upregulated proteins identified in lysates of MLN4924-treated ECs after data analysis, as depicted in Figure 1B. Proteins are ordered by increasing *p*-value. Fold change 10,000 denotes uniquely expressed proteins. * According to UniProt.

Protein Abbreviation	Protein Name	UniProt ID	Associated with *	*p*-Value	Fold Change
RPRD1B	Regulation of nuclear pre-mRNA domain-containing protein 1B	Q9NQG5	transcription	0.0037	10,000
EPHB2	Ephrin type-B receptor 2	P29323	contact-dependent signaling	0.0048	10,000
SQSTM1	Sequestosome-1	Q13501	autophagy	0.0052	2.44
WASL	Neural Wiskott–Aldrich syndrome protein	O00401	actin cytoskeleton	0.0058	6.16
HMOX1	Heme oxygenase 1	P09601	heme cleavage	0.0093	2.07
RRM2	Ribonucleoside-diphosphate reductase subunit M2	P31350	DNA synthesis	0.011	10,000
ILKAP	Integrin-linked kinase-associated serine/threonine phosphatase 2C	Q9H0C8	cell adhesion	0.015	6.19
AKR1A1	Alcohol dehydrogenase (NADP(+))	P14550	metabolism	0.022	1.96
ENO3	Beta-enolase	P13929	glycolysis	0.023	7.25
AAMP	Angio-associated migratory cell protein	Q13685	cell migration	0.023	7.24
MPI	Mannose-6-phosphate isomerase	P34949	glycosylation	0.023	7.12
HSPA4L	Heat shock 70 kDa protein 4L	O95757	molecular chaperone	0.023	3.38
CLASP1	CLIP-associating protein 1	Q7Z460	microtubules	0.025	3.36
CASP7	Caspase-7; Caspase-7 subunit p20; Caspase-7 subunit p11	P55210	cell death	0.029	10,000
OLA1	Obg-like ATPase 1	Q9NTK5	ATP hydrolysis	0.036	1.72
ARF3;ARF1	ADP-ribosylation factor 3; ADP-ribosylation factor 1	P84077;P61204	protein trafficking	0.038	1.83
GCLM	Glutamate-cysteine ligase regulatory subunit	P48507	glutathione biosynthesis	0.044	3.48
WDR5	WD repeat-containing protein 5	P61964	histone modification	0.045	6.25
MTSS1	Metastasis suppressor protein 1	O43312	actin cytoskeleton	0.045	6.22
CDKAL1	Threonylcarbamoyladenosine tRNA methylthiotransferase	Q5VV42	tRNA processing	0.045	6.10
CPNE2	Copine-2	Q96FN4	calcium-mediated signaling	0.045	6.09
ACSS2	Acetyl-coenzyme A synthetase, cytoplasmic	Q9NR19	acetylation	0.045	6.01
EXOC2	Exocyst complex component 2	Q96KP1	vesicular trafficking	0.045	6.38
DNAJC5	DnaJ homolog subfamily C member 5	Q9H3Z4	molecular chaperone	0.047	4.22
KRT83;KRT87P	Keratin, type II cuticular Hb3; Putative keratin-87 protein	P78385; A6NCN2		0.048	10,000
KRT86	Keratin, type II cuticular Hb6	O43790		0.050	10,000

**Table 4 cells-13-01609-t004:** The 6 significantly downregulated proteins identified in lysates of MLN4924-treated ECs after data analysis, as depicted in Figure 1B. Proteins are ordered by increasing *p*-value. * According to UniProt.

Protein Abbreviation	Protein Name	UniProt ID	Associated with *	*p*-Value	Fold Change
PDLIM4	PDZ and LIM domain protein 4	P50479	actin cytoskeleton	0.017	−8.16
MYO1C	Unconventional myosin-Ic	O00159	motor protein	0.018	−1.52
UPF1	Regulator of nonsense transcripts 1	Q92900	mRNA regulation	0.021	−1.73
CYC1	Cytochrome c1, heme protein, mitochondrial	P08574	oxidative phosphorylation	0.028	−7.02
DIAPH2	Protein diaphanous homolog 2	O60879	endosome dynamics?	0.028	−6.22
YTHDF2	YTH domain-containing family protein 2	Q9Y5A9	mRNA regulation	0.036	−3.54

## Data Availability

The mass spectrometry proteomics data have been deposited to the ProteomeXchange Consortium via the PRIDE [21] partner repository with the dataset identifier PXD053910.

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
