# Peer review of "AAMP and MTSS1 Are Novel Negative Regulators of Endothelial Barrier Function Identified in a Proteomics Screen"

_cells, 2024, doi:10.3390/cells13191609_

Round 1
Reviewer 1 Report
Comments and Suggestions for Authors
In this manuscript, Podieh et al used a proteomics approach to identify proteins differentially expressed after pharmacological inhibition of protein ubiquitination. The authors followed up on several actin-regulating proteins and showed that knockdown of AAMP or MTSS1 leads to modest TEER increases. Further work linked AAMP to the regulation of RhoB, cell contractility and membrane ruffles, providing a possible explanation for its effect on barrier function. As such this is a potentially important follow up to prior publications from the group.
The paper is intriguing, and uncovers two novel potential regulators of endothelial barrier function. However, the manuscript has several weaknesses that diminish the enthusiasm in its present form:
AAMP K/d appears to induce faster, but not higher, TEER. The increase in TEER after AAMP k/d is limited to the first 48h. Why is that? Is it possible that this protein may limit the formation of de novo junctions, but be dispensable for junction maintenance?
Faster area coverage could also explain faster TEER. A calcium switch experiment to determine barrier recovery could help assess these alternative explanations.
While direct ubiquitination and degradation of these two proteins is a likely explanation, alternative explanations exist (e.g., indirect increased gene expression, mRNA stability, or faster protein translation). To demonstrate the differences in protein stability the CHX experiments should be performed in cells pretreated or not with these E1 and E3 ligase inhibitors. Do they increase AAMP or MTSS1 half life?
To confirm an effect of AAMP on RhoA-GTP, please perform additional independent experiments and outlier analysis. The statistics on the data presented in figure 3 do not allow to conclude that “Taken together, siRNA-mediated loss of AAMP reduces the ability of RhoA and RhoB to induce cellular contraction”. Moreover, these experiments show a correlation, but not necessarily causality. How would a reduction in Rho activity lead to an increase in pMLC? How would that lead to reduced contraction?
It would be important to understand if these proteins have a role (i.e., k/d and lentiviral OE) in regulating barrier function in response to stress fiber-inducing agents (TNF, thrombin).
The data on wound healing is conflicting – why one assay show a difference but the other one does not? Is it possible that the ECIS assay did not completely kill the cells in the electrode? Is there a difference in survival of the AAMP k/d cells that could explain that conundrum?
Reviewer 2 Report
Comments and Suggestions for Authors
The manuscript by Podieh and colleagues, titled "AAMP and MTSS1 are novel negative regulators of endothelial barrier function identified in a proteomics screen," outlines the use of mass spectrometry screening to identify regulators of endothelial barrier function in HUVEC cells. The mass spectrometry screening focused on identifying proteins that are differentially expressed in HUVEC cells treated with ubiquitination inhibitors. After this, the authors conducted an siRNA screen for selected hits from the mass spectrometry data to identify new regulators of endothelial barrier function. AAMP and MTSS1 were identified as regulators, with a specific focus on AAMP. While this study represents a strong start to the investigation, it may lack novelty, considering that AAMP had already been identified previously. It would be beneficial to include a more in-depth analysis of the role of ubiquitination in this process.Some comments below are to be considered by the authors.
Major comments.
1. To identify proteins directly associated with RhoB, it would be more effective to conduct a proximity-labelling MS analysis.
2. The authors have identified AAMP as being upregulated upon MLN4924 treatment. It would be interesting and novel to demonstrate that AAMP is a target for CRLs. Currently, the only known PTM on AAMP is phosphorylation. It would be ideal to understand how AAMP is modulated by ubiquitination, especially considering that its expression is a poor prognosis marker for several cancers. Understanding its modulations and partners may open new avenues for cancer research.
3. In order to fully demonstrate that ubiquitination modulates AAMP and MTSS1 turnover, it would be useful to show that they are stabilized in CHX experiment supplemented with proteasome inhibitor or E1 inhibitor.
4. Does the localization of AAMP to cortical F-actin and cortactin depend on Rho signalling?
5. IP-MS would be a technique to identify potential interactors of AAMP (and MTSS1)
6. Figure 3D: The authors should confirm if the highest RohA.GTP value is not an outlier. More biological replicates might be required.
7. Figure 5B: is the location of AAMP affected by inhibition of ubiquitination?
8. Most figures lack internal controls. This would be essential, notably for experiments showing no significant difference.
Minor comments:
1. Figure 5B: the label is missing on the figure.
2. Figure 5C, D: it would be useful to see the location of the line used to plot the intensity profiles.
Comments on the Quality of English LanguageOverall language is fine. Some minor typos or syntax that may be reviewed as the sentence "A myc-co-IP showed, that cortactin and AAMP do not bind 497 directly, unlike coronin1B and cortactin" -> the comma after "showed" is not necessary. Also, the sentence is a little confusing on the first read.
Reviewer 3 Report
Comments and Suggestions for Authors
This study describes the role of rapid turnover of angio-associated migratory cell protein (AAMP) and metastasis suppressor protein 1 (MTSS1) as novel endothelial barrier regulatory proteins. The authors used the proteomics approach to identify short half-life hits, then applied literature search and tested the potential endothelial barrier-associated hits in experiments in vitro in human umbilical vein endothelial cells (HUVEC). The role of AAMP is tested in more detail. The manuscript is consistent and well-written. However, there are several issues to be addressed by the authors.
Major
1. Cell area measurements are not convincing. Firstly, I strongly assert, that the presented image of control HUVEC (Fig. 3C) seems very odd and does not match a common shape of single HUVEC; in contrast, siAAMP-treated cells look absolutely normal. Secondly, there is no description of the method used to measure cell area. Thirdly, 8-19 cells per condition is not a sufficient amount for measurements of this kind; at least 50 cells per condition should be analyzed. Lastly, changes in Rac1 and Cdc42 (manuscript lanes 417-419) do not support cell spreading. Collectively, the claim on the cell area change is weak and should be either withdrawn/made less tough or supported by increasing the amount of analyzed cells and presenting images that show several cells in one field of view.
2. There is a discrepancy between data presented in Fig. 3H and Fig. 3I. Without taking into account the semiquantitative nature of immunofluorescent staining, changes in pMLC fluorescence intensity should correspond well with the western blot data. If pMLC level increases locally, as suggested by the authors, in order the total pMLC level remains the same, there should have been a redistribution of pMLC fluorescence intensity. Surprisingly, the authors measured the total field of view fluorescence intensity and found a statistically significant increase. Please comment.
Moreover, an increase in pMLC is usually associated with endothelial cell contraction and decreased barrier function in contrast to the author’s conclusions (lanes 439-440). Additionally, RhoA-driven MLC phosphorylation involves Rho-associated protein kinase (ROCK), which readily induces MLC di-phosphorylation. Both mono- and di-phosphorylated MLC indicate active myosin II. The authors used an anti-pMLC antibody, which recognizes mono-pMLC only. More intense signal reported by the anti-pMLC antibody may reflect both de novo mono-phosphorylation (myosin activation) or dephosphorylation of the di-phosphorylated form (less active myosin). Since no control was done for di-pMLC, caution should be exercised when interpreting pMLC data. Please consider.
3. The same cell migration process provides for wound closure as monitored by ECIS as well as in the scratch assay. However, only the ECIS-based measurement revealed some difference. Please comment. Additionally, electric wounding tends to vary significantly. From the Methods section or the Fig. 4 legend, it is unclear how many ECIS-based wounding experiments were performed. What does n=3 mean in the Fig 4 legend? Were there three wells/electrodes in one experiment or three independent experiments? Please clarify in the Methods and add the ECIS data obtained in other wound healing experiments in the Supplementary Materials or at least in the Supporting data.
Minor
1. In the proteomics analysis with MLN4924, RhoB was not identified as an upregulated protein (Fig. S1D). In contrast, Western blots in Fig. S2C suggest RhoB is upregulated after treatment with MLN4924. Please, comment in Discussion on this discrepancy as it is a potential weakness of the approach to identify important players like RhoB, which is used as a positive control in the study.
2. Coomassie staining is a semiquantitative approach to ensure equal protein amounts between lysates. At least a densitometric analysis should be applied to compare lanes. Measurement of total protein in samples using BCA, Bredford, Lowry, etc. methods is more relevant.
3. In methods, please indicate the final DMSO concentration (%) used in experiments, because DMSO is known to affect protein phosphorylation (signaling) even at low doses.
4. Table 1 to 4 captions read ‘Fc 1000 denotes…’. Firstly, does Fc stand for fold change? Please disclose the acronym. Secondly, in the tables, a value of 10000,00 is indicated for unique proteins. By the way, a separate Abbreviation section would be helpful to identify acronyms frequently found in the text.
5. Lane 580, please add numbering to the cited references to make the text reader-friendly.
Reviewer 4 Report
Comments and Suggestions for Authors
The authors’ study is devoted to investigation of the new players in regulation of vascular endothelium permeability. This process is participating in a lot of pathophisiological conditions, such as inflammation, atherosclerosis, infections, tumor growth etc. Basically, for investigating the hypothesis non-driven molecular players, people use the differential (particularly, in treating vs control experiments) transcriptomic, proteomic, metabolomic approaches, and then they pick some of most promisable targets and check them in subsequent studies. But in this study, the authors investigate the (patho-)physiological role of the short living proteins. So, they used not frequently applying ubiquitinome approach, which, for my oppinon, gave some promisable results. The main result of this study is discovering of AAMP and MTSS as the short-living proteins, up-regulating of endothelial permeability. For me, the mechanism of there action is vague, but, surely, it can be investigated in future research. Although the results can be published, I have to address some major and minor points for the impoving the manuscript:
Major:
1. Why the RhoB content according to the proteome data didn’t increase in case of Cullin E3 ligase inhibition (Fig S1D), whyle in the Western study it was increased by this inhibitor (Fig. S2 C, D).
2. At the beginning the authors wrote (432) “Both RhoA.GTP and RhoB.GTP induce phosphorylation of myosin light chain (pMLC), leading to actomyosin-based contractility.” It means that phosphorylation of myosin light chain is a way of increasing of endothelium permeability”. But further they concluded (438), that “siRNA-mediated loss of AAMP reduces the ability of RhoA and RhoB to induce cellular contraction, leading to cell spreading, accompanied by increased pMLC, and resulting in an increase in endothelial barrier function.” So, in this case, phospho-MLC will result the increase in barrier endothelium function. So, phospho-MLC - a way of increasing or decreasing of endothelial permeability??? It should be pointed clear.
3. Section 3.4. Actually, I don’t follow the significanse of this data. Why the migration of EC? The authors studied the barrier function of endothelium. And it seems that these data are not new (line 466). May be it should be shown in supplementary, as an additional measure of functional significance of AAMP knockdown? And two phrases are inconsistent “Consistent with previous findings, depletion of AAMP lead to a delayed wound closure compared to N.T. control in a wound healing assay performed with ECIS (Figure 4A)” and “Conversely, silencing of AAMP did not affect the closure of a scratch in a HUVEC monolayer (Figure S4A)”.
4. Line 118 Misprint HVUECs
5. Fig. 5. Letter B is missing.
Round 2
Reviewer 1 Report
Comments and Suggestions for Authors
The authors have successfully addressed all prior concerns
Reviewer 2 Report
Comments and Suggestions for Authors
The authors have satisfactorily answered all my questions.
Reviewer 3 Report
Comments and Suggestions for Authors
The authors have provided thorough and careful responses and amended the manuscript in accordance with round 1 review comments. No more concerns.